# No Role of Osteocytic Osteolysis in the Development and Recovery of the Bone Phenotype Induced by Severe Secondary Hyperparathyroidism in Vitamin D Receptor Deficient Mice

**DOI:** 10.3390/ijms21217989

**Published:** 2020-10-27

**Authors:** Barbara M. Misof, Stéphane Blouin, Jochen G. Hofstaetter, Paul Roschger, Jochen Zwerina, Reinhold G. Erben

**Affiliations:** 1Ludwig Boltzmann Institute of Osteology at the Hanusch Hospital of OEGK and AUVA Trauma Centre Meidling, 1st Medical Deptartment, Hanusch Hospital, 11140 Vienna, Austria; stephane.blouin@osteologie.lbg.ac.at (S.B.); jochen.hofstaetter@oss.at (J.G.H.); paul.roschger@osteologie.lbg.ac.at (P.R.); jochen.zwerina@osteologie.lbg.ac.at (J.Z.); 2Michael Ogon Laboratory for Orthopaedic Research, Orthopaedic Hospital Vienna Speising, 1130 Vienna, Austria; 3Department of Biomedical Sciences, University of Veterinary Medicine, 1210 Vienna, Austria; Reinhold.Erben@vetmeduni.ac.at

**Keywords:** vitamin D receptor, mice with a non-functioning vitamin D receptor, bone mineralization density distribution, osteocyte lacunae sections, quantitative backscattered electron imaging, osteocytic osteolysis

## Abstract

Osteocytic osteolysis/perilacunar remodeling is thought to contribute to the maintenance of mineral homeostasis. Here, we utilized a reversible, adult-onset model of secondary hyperparathyroidism to study femoral bone mineralization density distribution (BMDD) and osteocyte lacunae sections (OLS) based on quantitative backscattered electron imaging. Male mice with a non-functioning vitamin D receptor (VDR^Δ/Δ^) or wild-type mice were exposed to a rescue diet (RD) (baseline) and subsequently to a low calcium challenge diet (CD). Thereafter, VDR^Δ/Δ^ mice received either the CD, a normal diet (ND), or the RD. At baseline, BMDD and OLS characteristics were similar in VDR^Δ/Δ^ and wild-type mice. The CD induced large cortical pores, osteomalacia, and a reduced epiphyseal average degree of mineralization in the VDR^Δ/Δ^ mice relative to the baseline (−9.5%, *p* < 0.05 after two months and −10.3%, *p* < 0.01 after five months of the CD). Switching VDR^Δ/Δ^ mice on the CD back to the RD fully restored BMDD to baseline values. However, OLS remained unchanged in all groups of mice, independent of diet. We conclude that adult VDR^Δ/Δ^ animals on an RD lack any skeletal abnormalities, suggesting that VDR signaling is dispensable for normal bone mineralization as long as mineral homeostasis is normal. Our findings also indicate that VDR^Δ/Δ^ mice attempt to correct a calcium challenge by enhanced osteoclastic resorption rather than by osteocytic osteolysis.

## 1. Introduction

The action of the active form of vitamin D, 1,25(OH)_2_D, is mediated by the vitamin D receptor (VDR) and plays an important role for calcium homeostasis [1,2,3,4,5]. Vitamin D stimulates intestinal calcium transport to maintain normal serum calcium levels, which, in turn, provides calcium supply to bone. The dietary calcium content is critical for calcium homeostasis. At a normal-to-low calcium diet, the active calcium transport controlled by 1,25(OH)_2_D predominates in the intestine, whereas at a high calcium intake, calcium is absorbed via passive diffusion [2]. In the former case, when intestinal calcium transport is inadequate, the release of calcium from bone is enhanced via the stimulation of the parathyroid gland to increase parathyroid hormone (PTH) secretion. PTH increases osteoclastic bone resorption while simultaneously raising 1,25(OH)_2_D blood levels that, in turn, suppresses bone matrix mineralization [1]. In addition, evidence from lactating mice, as well as from many other experimental models, has suggested the existence of a very fast mechanism (within minutes) to release or bind calcium from the inner surface of the osteocytic lacunar canalicular network in bone [6,7]. In lactating mice, this physiologic condition of perilacunar remodeling/osteocytic osteolysis is associated with elevated circulating parathyroid hormone related protein (PTHrP) levels. PTHrP and PTH bind with similar affinities to the PTHR1 receptor, which is also expressed in osteocytes [6]. Evidence of osteocytic osteolysis in hyperparathyroidism was reviewed in [8], pointing toward a role of elevated PTHrP or PTH levels and to a role of osteocytic osteolysis in calcium retrieval in pathologic conditions.

Global VDR^-/-^ mice, as well as mutants with a non-functioning VDR (VDR^Δ/Δ^), are characterized by an intestinal calcium absorption defect and develop severe secondary hyperparathyroidism (sHPT), an increased bone turnover, and a decreased bone mineral density, as well as concomitant skeletal abnormalities consistent with rickets including increased osteoid volume, cortical thinning and growth plate abnormalities during few weeks postnatally when kept on a normal mouse chow [9,10,11]. Moreover, Lieben et al. showed that in mouse mutants with a specific VDR inactivation in the intestine and in osteocytes, the osteoblastic VDR signaling suppressed calcium incorporation in bone by directly stimulating the transcription of genes encoding mineralization inhibitors [12]. Intestine-specific VDR knockout mice were characterized by an increased bone turnover, decreased bone volume, mineralization defects, and hyperosteoidosis surrounding osteocytes, all likely to ensure normal levels of calcium in the serum [12].

Feeding global VDR^−/−^ or VDR^Δ/Δ^ mice with a calcium/phosphate/lactose-enriched, so-called rescue diet from the beginning of their lives prevents them from hyperparathyroidism and skeletal abnormalities [9,13,14]. Hence, although the VDR is widely expressed in bone cells, it appears to lack a physiological function under a normal calcium supply [9,15]. In the current study, we sought to harness the VDR^Δ/Δ^ mice mouse model to induce a controlled and reversible state of severe sHPT in adult, non-growing mice. We reported previously that VDR^Δ/Δ^ mice maintained on the rescue diet until adulthood showed an indistinguishable bone phenotype relative to wildtype (WT) mice but developed severe sHPT when they were subsequently fed a low calcium challenge diet, and that this adult-onset sHPT could be completely corrected by the administration of the rescue diet [9]. Using this model of reversible adult-onset sHPT, we sought to answer the questions of whether bone mineralization density distribution (BMDD) assessed by quantitative backscattered electron imaging (qBEI) is normal in VDR deficient mice and whether osteocytic osteolysis in non-growing mice occurs in the absence of a functioning VDR. 

## 2. Results

### 2.1. Strong Correlations of Serum PTH, ALP, and Serum Calcium Levels with Bone Matrix Mineralization, but Not with Osteocyte Lacunar Size 

The relationship of BMDD and osteocyte lacunae sections (OLS) outcomes with previously measured serum PTH, serum calcium, and total alkaline phosphatase (ALP) from identical animals showed strong monotonic associations between these serum biochemical measures with the average Ca concentration of the mineralized bone area (CaMean) (cortical bone (Ct.) and epiphyseal spongiosa (Es.)) and the amount of tissue area mineralized below 17.68 wt% Ca (CaLow) (Ct., Es., and metaphyseal spongiosa (Ms.)) (Spearman rank order (SRO) correlation coefficients are given in Table 1). No significant correlation was found between BMDD outcomes and serum phosphorus. Interestingly, none of the correlations between OLS characteristics and biochemical outcomes achieved statistical significance.

### 2.2. Bone Phenotype of VDR^Δ/Δ^ Mice Is Similar to That of WT at Baseline but Significantly Different after 2 Months CD 

At baseline, no significant difference in BMDD outcomes (in Ct., Es., or Ms. femoral bone), OLS parameters, or the microstructure of mineralized bone could be observed between the two genotypes (all comparisons VDR^Δ/Δ^ vs. WT were *p* > 0.05; Figure 1 and Table 2). Furthermore, no extensive presence of osteoid (based on qualitative observation in inverted qBEI images—not shown) could be detected. Thus, the rescue diet (RD) completely prevented abnormalities in mineralization, osteocyte lacunae, and microarchitecture in four-month-old VDR^Δ/Δ^ mice. 

The feeding of the calcium challenge diet (CD) for two months had differential effects on the BMDD in the WT mice and VDR mutants, i.e., we found a significant interaction between genotype and diet by a two-way ANOVA. Consequently, WT and VDR^Δ/Δ^ groups were compared separately for diet effects. Within VDR^Δ/Δ^ animals, Es.CaMean (−9.5%), Ms.CaMean (−10.4%), and Ms.CaPeak (−6.9%) were decreased, whereas Ct.CaLow (+89%), Es.CaLow (+200%), and Ms.CaLow (+ 95%, all *p* < 0.001) were highly increased after two months with the CD compared to the baseline. In contrast, there was only a small change in Ms.CaMean, whereas all other aforementioned parameters remained at the baseline level after two months with the CD in WT mice. These differential effects of diet led to significant differences between WT and VDR^Δ/Δ^ after two months with the CD, including significantly lower Es.CaMean and Ms.CaMean values, as well as higher Ct.CaLow, Es.CaLow, and Ms.CaLow values in VDR^Δ/Δ^ animals compared to the WT animals (see Figure 1). 

All other BMDD outcomes (including those where the interaction term was not significant) are shown in Figure 1. The factor genotype in the two-way ANOVA did not yield significant differences for BMDD parameters, but it was significant for the OLS area, which was generally lower in the VDR^Δ/Δ^ versus WT mice (−12.1%, *p* = 0.033). All other OLS characteristics were not significantly different based on the two-way ANOVA comparison. 

An analysis of mineralized bone structure in VDR^Δ/Δ^ after two months with the CD showed significantly lower md.BV/TV (−70%), md.Tb.Th (−36%, both *p* < 0.01), and md.Tb.N (−50%, *p* < 0.05) in the epiphyseal spongiosa but not in metaphyseal bone compared to the WT mice. Additionally, large pores were observed in the cortical midshaft from the VDR^Δ/Δ^ animals. These pores were partly filled with abnormally low mineralized bone matrix that contained roundish osteocyte lacunae sections of larger size (as shown for the nine-month-old animals in the following subsection), while average OLS characteristics of the entire bone area were not significantly changed, as mentioned above.

### 2.3. From the 9-Month-Old VDR^Δ/Δ^ Animals, Only Those on RD Have Similar BMDD Compared to Baseline 

ANOVA or ANOVA on ranks analysis and a subsequent post-hoc analysis revealed significant differences for BMDD parameters among the nine-month-old VDR^Δ/Δ^ groups compared to the baseline. The BMDD and OLS results are summarized in Figure 2. Ct.CaLow and Ms.CaLow values were both higher in the VDR^Δ/Δ^ 5CD mice compared to the baseline (+164% and +108%, respectively, both *p* < 0.001). Es.CaMean was decreased in the VDR^Δ/Δ^ 5CD mice compared to the baseline (−10.3%, *p* < 0.01), as well as compared to the VDR^Δ/Δ^ 5RD mice (*p* < 0.01). Similarly, Es.CaMean was also decreased in the VDR^Δ/Δ^ 5ND (normal diet) mice compared to the baseline (−6.4%, *p* < 0.05), as well as to the VDR^Δ/Δ^ 5RD mice (*p* < 0.05). VDR^Δ/Δ^ 5RD was the only group among the nine-month-old animals that did not differ in any BMDD parameter compared to the baseline. Furthermore, the average OLS characteristics were not significantly different among the nine-month-old VDR^Δ/Δ^ diet groups (Figure 2). 

As already observed in the VDR mutants after two months of the CD, all nine-month-old mouse groups revealed large cortical pores in the midshaft diaphysis, independent of the type of diet. In the VDR^Δ/Δ^ animals on the CD or the ND, these large cortical pores included osteoid and abnormally low mineralized bone matrixes with OLS of larger size and roundish shape (an example is shown in Figure 3). In addition to these observations, extended osteoid areas were found also in epiphyseal and metaphyseal spongiosa in the VDR^Δ/Δ^ 5CD and VDR^Δ/Δ^ 5ND groups, as shown in Figure 3 for the metaphysis. In the VDR mutants after three months of the RD, such abnormally low mineralized or unmineralized bone was not observed in either the cortical or cancellous bone.

## 3. Discussion

In the present work, the change in bone phenotype relating to previously obtained biochemical measures (serum calcium and PTH, in particular) was studied in an adult-onset sHPT model in mice with a non-functioning VDR. Our correlation analyses indicated an association of serum calcium, PTH, and ALP with the average degree of calcium concentration and the percentage of low mineralized bone areas in all studied bone compartments, which is in line with the well-known dependency of the BMDD on bone turnover/formation in patients with primary or secondary hyperparathyroidism [16,17,18]. However, the lack of any relationship between serum PTH and osteocyte lacunae characteristics was unexpected. In general, our BMDD findings perfectly fit to what is already known about VDR knockout mice, which is that the rescue diet is able to maintain normal mineralization and that the challenge diet decreases average bone matrix mineralization. Additionally, the decrease in bone volume and increase in cortical pores is known from previous works. Altogether, this shows how the sHPT affects bone turnover and mineralization. What was unknown until now is that the rescue diet is also able to restore bone matrix mineralization in adult VDR^Δ/Δ^ mice and that neither the challenge nor rescue diets have a significant influence on osteocyte lacunae characteristics.

An important prerequisite for our study of dietary effects was that four-month-old VDR^Δ/Δ^ animals maintained on the RD did not show skeletal abnormalities, thus providing normal baseline levels of all measured bone characteristics including BMDD, OLS, and microarchitecture. This finding is in agreement with normal values for serum Ca and phosphorus and only slightly elevated PTH levels in four-month-old VDR^Δ/Δ^ mice on the RD [9], and it is also in line with previous observations of VDR knockout mice fed with the rescue diet [14]. The RD prevented the development of osteomalacia in other VDR knockout mice [19], and, moreover, no signs of osteomalacia were observed previously in our RD-fed VDR^Δ/Δ^ mice at the baseline [9], which was confirmed in the current study by the absence of enhanced amounts of unmineralized bone matrix, as visualized by inverted qBEI imaging. Taken together, these findings suggest that the RD is able to maintain a normal bone phenotype in adult VDR^Δ/Δ^ mice. Thus, our model enabled us to study the effects of the challenge diet in adult VDR^Δ/Δ^ mice independent of growth effects. It is noteworthy that no information about possible premature bone ageing in the VDR mutants is available. Some evidence of premature ageing was found based on effects in lipid and carbohydrate metabolism, although the VDR^Δ/Δ^ mice might not be an appropriate model for studying such effects because the older VDR mutants have a lean phenotype [20]. Moreover, there is no evidence that BMDD changes occur by ageing adult wild-type mice to a much older age than studied in the present work [21,22]. This suggests that we were able to observe dietary effects only, and we can generally rule out any significant influence of age on BMDD in the present work. This was likely also the case for OLS characteristics where no age-differences were observed in mice between 6 and 18 months [23].

The switching of the adult VDR^Δ/Δ^ mice on the RD to the CD caused severe sHPT and hypocalcemia [9], in addition to resulting in severe abnormalities in mineralization and microarchitecture; however, it did not induce any alterations in OLS characteristics compared to WT and VDR^Δ/Δ^ baseline animals. In line with the sHPT, we found cortical tunneling resorption in the VDR^Δ/Δ^ mice, confirming the previously reported loss of cortical BMD as measured by pQCT caused by the CD [9]. It is noteworthy that intracortical remodeling is normally absent in murine bone. Furthermore, the mineralized bone volume in the epiphyseal spongiosa was decreased by the CD in VDR^Δ/Δ^ animals. Both changes in bone phenotype described above are compatible with the occurrence of sHPT leading to augmented bone resorption, as well as to increased new bone formation, resulting predominantly in osteoid and/or a bone matrix with a distinctly lower degree of mineralization (as reflected by an increased CaLow and accordingly reduced CaMean). 

In contrast to WT mice, which remain normocalcemic on the CD, VDR^Δ/Δ^ mice on the CD suffer from severe hypocalcemia [9]. In dietary calcium deficiency or other conditions where calcium is needed, two mechanisms for calcium retrieval might be activated—one by osteoclastic resorption and the other by osteocytic osteolysis [6,8,16,24,25,26]. Osteocytic osteolysis might result in a rapid calcium release from the mineralized bone matrix of the mineral around osteocytic lacunae, which would consequently increase the area of the sectioned lacunae in qBEI images, as was demonstrated in lactating mice [6]. Its link to PTHrP and PTH levels and its dependency on the PTHR1 receptor led to the hypothesis that osteocytic osteolysis might be stimulated in pathologic conditions with increased PTH. Indeed, osteocytic osteolysis has been found in rats treated with PTH [24], as well as in patients with sHPT [16,25]. Furthermore, an increased perilacunar area was also reported in growing mice with global [14] or targeted ablation of the VDR [12], as well as in WT mice after dietary calcium restriction after weaning [12]. Surprisingly, in contrast to these observations, changes in OLS properties were absent in our VDR^Δ/Δ^ mutants that had dramatically increased PTH levels up to 45- and 65-fold of the baseline levels after two and five months of the CD, respectively [9]. It is unclear whether the adult-onset experimental setup in non-growing mice, dietary differences (compared to the previous study [12], our challenge diet had a somewhat higher Ca and lower phosphate content), or the inactivation of the VDR in our mouse model blunted osteocytic osteolysis. This cannot be definitely answered here because we were also not able to observe dietary OLS changes in the WT mice. It is noteworthy that the WT animals on the CD were neither hypocalcemic nor did they reveal any signs of defective mineralization [9]. Thus, it is likely that a much lower dietary calcium intake would be necessary to challenge any mechanism for calcium release from bones in WT animals. 

Collectively, our findings indicate that four-month-old global VDR^Δ/Δ^ mice aim to correct low serum calcium levels by the activation of osteoclastic resorption rather than by osteocytic osteolysis. Though we are not able to rule out that demineralization processes (independent of the osteocytic activities) might also contribute to the observed cortical pores and defects in mineralization, there is evidence that osteoclastic resorption is responsible for the observed features. The qBEI and inverted qBEI images clearly demonstrated eroded surfaces corresponding to resorption lacunae within the pores and that the pores were only partly filled with osteoid. Both would not be seen in a general demineralization of bone and point toward osteoclastic resorption as the fundamental process leading to the large pores. It is noteworthy that the increased urinary deoxypyridinoline/creatinine excretion reported in the previous work by Weber et al. [9] supported this assumption. 

Interestingly, the newly formed bone matrix that can be found occasionally within these pores was not getting adequately mineralized and locally exhibited OLS of larger size. Given their local occurrence within the pores, it is likely that these larger lacunae were due to the defective mineralization of the newly formed bone during the CD phase. We also suppose that the observed thick osteoid layer covering the mineralized trabeculae in epiphyseal and metaphyseal bone originated from the defective mineralization of newly formed bone and not from demineralization. In particular, the increased alkaline phosphatase, as observed in the previous work [9], points toward relatively high bone formation followed by defective mineralization.

The finding of large cortical pores together with the reported low serum calcium levels in the VDR^Δ/Δ^ mice on the CD [9] indicate that the high osteoclastic resorption activity was not able to fully correct the hypocalcemia. Lieben et al. reported normocalcemia and concurrently decreased bone matrix mineralization in mice with a targeted ablation of the VDR in the intestine, a finding that suggested that the maintenance of normocalcemia has priority over skeletal integrity [12]. In contrast, in our mice, we found decreased bone matrix mineralization associated with hypocalcemia. 

When the VDR^Δ/Δ^ mice with established sHPT after two months of the CD were fed the ND or the RD for another three months, only those on the RD had completely restored BMDD at all measured compartments, i.e., were similar to VDR^Δ/Δ^ at the baseline. It is noteworthy that the latter group of mice was also the only group of VDR mutants in which serum calcium and PTH levels returned to baseline levels [9]. However, the large cortical pores described for the VDR^Δ/Δ^ 2CD mice remained in all nine-month-old VDR^Δ/Δ^ groups. Moreover, neither the continuation of the CD for further a three months nor feeding with the ND or the RD affected the average osteocyte lacunar properties of the entire diaphyseal cortical compartment in the VDR^Δ/Δ^ mice. 

Our study had several limitations. The relatively low number of mice per group may have prevented us from observing very subtle dietary effects. Furthermore, OLS could only be studied in cortical bone because the bone volume of the epiphyseal or metaphyseal spongiosa was too low to provide adequate OLS numbers for representative values for OLS characteristics. Another important limitation for the interpretation of our OLS analysis was the lack of a positive control, consisting of an adult-onset group of WT animals on a challenging low calcium diet, as mentioned above. Moreover, our exclusive use of male mice might have prevented us from observing possible sex-related differences in bone metabolism regulation by the VDR. Sex-related differences in bone microstructure and formation have been reported for VDR knockout mice fed with the RD compared to control animals [19]. 

We conclude that adult four-month-old VDR^Δ/Δ^ animals on the RD lack any abnormalities in mineralization, microarchitecture, or OLS, thus further corroborating the notion that VDR signaling is not essential for normal bone metabolism and mineralization as long as mineral homeostasis is normal. Switching these mice to a CD caused abnormalities in bone matrix mineralization and a decrease in bone volume, in line with the onset of sHPT. The appearance of large cortical pores together with unchanged OLS properties suggest that VDR^Δ/Δ^ mice on a CD attempt to maintain serum calcium levels by enhanced osteoclastic resorption rather than by osteocytic osteolysis. The reversal of the sHPT and hypocalcemia by switching VDR^Δ/Δ^ mice on the CD back to the RD fully corrected the abnormal BMDD and the defective mineralization (osteoid or abnormally low mineralization) in cortical pores and the metaphyseal spongiosa. 

## 4. Materials and Methods 

### 4.1. Bone Samples

We studied distal femora from a total of 33 male mice at different ages and different genotypes (*n* = 11 WT and *n* = 22 homozygous VDR^Δ/Δ^). These femora were harvested from mice that were primarily studied for the function and proliferation of parathyroid gland cells under different diets [9]. The bone serum parameters measured in a previous work [9] were used for correlation analyses with bone matrix mineralization and osteocyte lacunae characteristics outcomes in the current work. The generation of the VDR^Δ/Δ^ mice and the design of the present study using the adult onset of sHPT was described elsewhere [3,9]. The mice were fed with different types of diets: a challenge diet (CD; containing 0.5% calcium, 0.4% phosphorus, and 0% lactose), a normal diet (ND; 0.9% calcium, 0.7% phosphorus, and 0% lactose), and a rescue diet (RD; 2.0% calcium, 1.25% phosphorus, and 20% lactose). The VDR mutants studied in this work represented an adult-onset model of sHPT. Both groups of mice (VDR^Δ/Δ^ and WT) were fed with the RD until the age of 4 months (which represented the baseline groups *WT-BL* and *VDR*^Δ/Δ^
*BL*), which was able to keep serum PTH levels at nearly normal levels and prevented the development of osteomalacia in the VDR^Δ/Δ^ mice, as reported previously [9]. At 4 months of age, both WT and VDR mutants were switched to the CD, which is known to cause severe sHPT in the VDR^Δ/Δ^ animals [9]. WT and VDR^Δ/Δ^ bones were harvested after two months of CD (*WT-2CD* and *VDR*^Δ/Δ^
*2CD* groups). This was followed by a further three months of either the CD (*VDR*^Δ/Δ^
*5CD)*, the ND (*VDR*^Δ/Δ^
*5ND)*, or the RD in the VDR^Δ/Δ^ animals *(VDR*^Δ/Δ^
*5RD)*. These latter three study groups represented the nine-month-old animals, which represented the oldest mice studied in this work (the study design is also shown in Appendix A
Figure A1). All animal procedures were approved by the Ethical Committee of the University of Munich and the local government authorities (Az. 211-2531-53/98, extended on 23 -04-2002).

### 4.2. Quantitative Backscattered Electron Imaging (qBEI)

In qBEI, the electron signal backscattered from about 1.5 micron-thick surface-layer of the sectioned bone area was acquired (Figure 4A). This signal was proportional to the weight concentration of mineral (hydroxyapatite) and thus related to the weight% (wt%) calcium in bone, as described in detail previously [27],. In the present work, we used a scanning electron microscope (DSM 962, Zeiss, Oberkochen, Germany) equipped with a four quadrant semiconductor backscattered electron detector. Calibrated digital backscattered electron images with a 200× nominal magnification (resolution 0.88 microns/pixel) were acquired, from which gray-level histograms (frequency distributions, so-called bone mineralization density distributions or BMDDs) were obtained and indicated the percentage of mineralized bone area with a certain gray level (Figure 4B). The gray scale was calibrated using the “atomic number contrast” between carbon (C, *Z* = 6) and aluminum (Al, Z = 13) as reference materials. One gray level step corresponded to 0.17 wt% Ca as a consequence of this calibration protocol. 

BMDD measurements were performed in cortical bone (Ct) in the diaphysis of the mid-shaft region, as well as in epiphyseal (Es) and metaphyseal spongiosa (Ms). The following parameters were calculated from the Ms.BMDD, Es.BMDD, and Ct.BMDD of each sample: CaMean, which is the weighted average Ca concentration of the mineralized bone area; CaPeak, which is the mode/peak calcium concentration representing the most frequently observed Ca concentration; CaWidth, characterized as the width at half-maximum of the BMDD histogram peak indicating the heterogeneity of mineralization; CaLow, representing the amount of tissue area mineralized below 17.68 wt% Ca (corresponding to primary mineralization in human bone [28], indicated in Figure 4B); and CaHigh, which represents the percentage of highly mineralized bone areas (above the average 95th percentile of calcium concentrations of the WT-BL samples). 

### 4.3. Analysis of Structural Morphometric Parameters of Mineralized Bone Tissue

The calibrated digital images were additionally used for the analysis of the structural morphometric parameters/microarchitecture of the mineralized bone tissue in epiphyseal and metaphyseal spongiosa by discriminating between the grey-levels of mineralized and unmineralized tissue or embedding material. A custom-made automated image analysis routine (NIH Image 1.52, W. Rasband, National Institutes of Health), as described elsewhere [29], was applied to the qBEI images. Measurements of md.BV/TV, md.Tb.Th, and md.Tb.N (md. = mineralized) were performed in the sagittal sections of the epiphyseal spongiosa inside the epiphyseal cortical bone envelope and within the metaphyseal region at a distance of 500–1500 microns to the growth plate. Such analysis was performed exclusively in the WT-BL, VDR^Δ/Δ^ BL, WT-2CD, and VDR^Δ/Δ^ 2CD study groups, because only these groups allowed for direct conclusions about the separate effect of diet on microarchitecture excluding concurrent well-known changes on the microarchitecture in ageing mice [30].

Additionally, qBEI images with modified image contrast settings including a grey-scale inversion (white pixels became black and vice versa) were used to qualitatively evaluate the presence or absence of extended amounts of osteoid (Figure 3).

### 4.4. Osteocyte Lacunae Sections (OLS) Analysis

The OLS analysis in the present work was a 2D-characterization of osteocyte lacunae sections based on the qBEI images [31] of the longitudinal sections of femora. For this purpose, qBEI images were transformed to binary images using a threshold based on a fixed grey level (5.2 wt% Ca) (for details, see previous work [31]). These binary images were analyzed for OLS based on a custom-made macro in the ImageJ software (version 1.52n; NIH, Bethesda, MD, USA) [32]. Subsequently, the OLS were extracted using minimum and maximum size thresholds of 5 and 80 µm^2^, respectively. Five parameters were obtained:(1)$$OLS-density\;\left(\frac{nb.}{mm^{2}}\right)=\frac{number\;of\;OLS\;}{mm^{2}\;mineralized\;bone\;area}$$
(2)$$OLS-porosity\;\left(\%\right)=100\times \frac{OLS\;total\;area}{mineralized\;bone\;area+OLS\;total\;area}$$
(3)$$OLS-area\;\left(µm^{2}\right)=mean\;OLS\;area\;per\;sample$$
(4)OLS-perimeter (µm)=mean OLS perimeter per sample
(5)$$OLS-aspect\;ratio=OLS-AR=mean\;\frac{\mathrm{major}\;\mathrm{axis}}{\mathrm{minor}\;\mathrm{axis}}\;of\;the\;fitted\;ellipse\;for\;each\;OLS$$

OLS-AR = 1 indicates a circle, and increasing values indicate an increasingly elongated shape of the OLS. OLS-AR values > 10 were excluded. The presence and viability of the cells within the OLS could not be evaluated by this method. 

### 4.5. Statistical Analysis

SigmaStat Version 4.0 (Systat Software Inc., San Jose, CA, USA) was used for the following statistical analyses: (1) We correlated all BMDD or OLS data with biochemical outcomes based on Spearman rank order (SRO) correlation analysis. As a large number of correlation tests were analyzed, we considered *p* < 0.0006 (α/number of correlation tests = 0.05/80) for significance in order to avoid false-positive results due to multiple testing (thus, only these results are considered in Table 2). (2) We compared the bone phenotype of VDR^Δ/Δ^ mice with that of the WT mice at the baseline and after two months of the CD. This was analyzed based on a two-way ANOVA of BMDD and OLS outcomes using factor genotype (WT versus VDR^Δ/Δ^), factor diet (baseline versus two months with the CD), and the interaction of these factors. The mineralized microstructure was compared by a t-test or a Mann–Whitney rank sum test (for non-normally distributed data) at the baseline and after two months with the CD. (3) We compared the bone phenotype from the nine-month-old VDR^Δ/Δ^ animals (5CD, 5ND, and 5RD VDR^Δ/Δ^ mice) with that of the VDR^Δ/Δ^ mice at the baseline based on an ANOVA or ANOVA on ranks (for non-normally distributed data) comparison, followed by a post-hoc comparison (based on Holm–Sidak or Dunn’s Method, respectively). For all aforementioned comparisons, two-sided *p* < 0.05 was considered for significance. As not all data in VDR^Δ/Δ^ groups were normally distributed, median (25^th^ and 75^th^ percentiles) are shown in the figures and tables. 

## Figures and Tables

**Figure 1 ijms-21-07989-f001:**
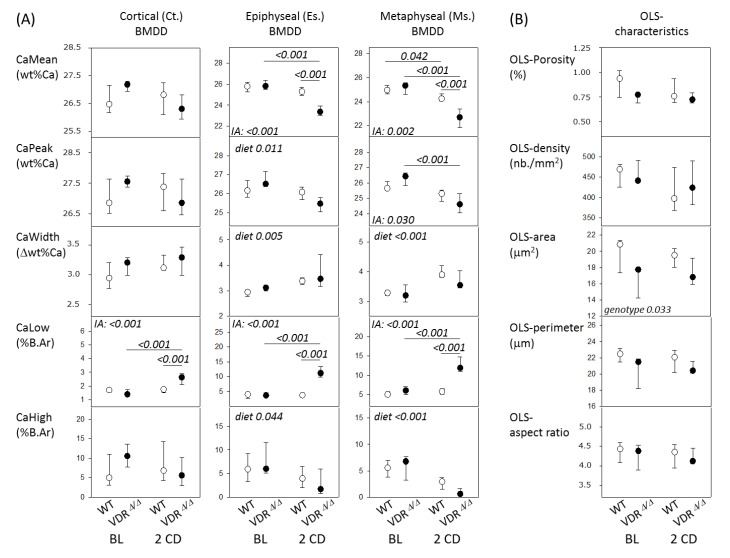
(**A**) BMDD results of cortical diaphyseal bone, epiphyseal, and metaphyseal spongiosa and (**B**) osteocyte lacunae sections (OLS)-characteristics from WT (open) and VDR mutants (full symbols). Data are median (25th and 75th percentiles). *p*-values according post-hoc analysis following a two way ANOVA are shown. IA = interaction between factors genotype and diet; BL = baseline; 2CD = after 2 months with the CD.

**Figure 2 ijms-21-07989-f002:**
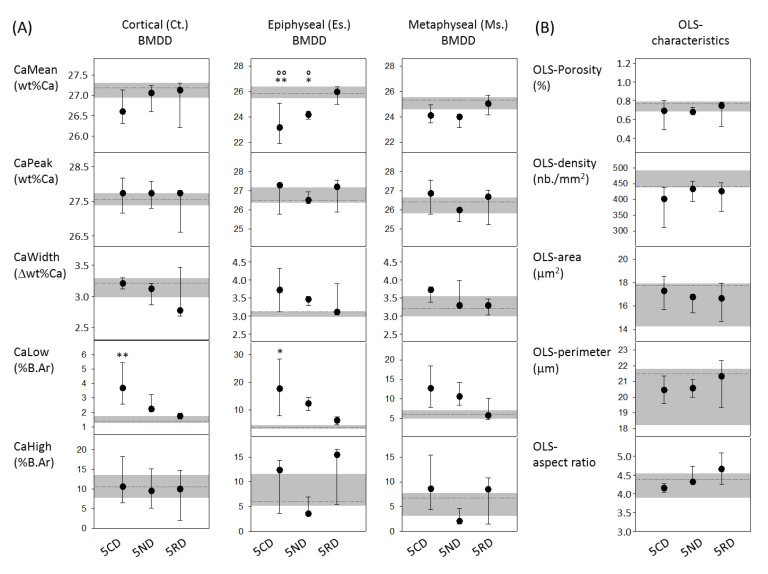
(**A**) BMDD results of cortical diaphyseal bone, epiphyseal, and metaphyseal spongiosa; (**B**) OLS-characteristics from 9-month-old VDR mutants. Data are median (25^th^ and 75^th^ percentiles) from VDR^Δ/Δ^ 5CD, VDR^Δ/Δ^ 5ND (normal diet), and VDR^Δ/Δ^ 5RD (rescue diet). The grey line and area in the background indicate the median and the 25th and 75th percentiles of the VDR^Δ/Δ^ at the baseline. ** *p* < 0.01, * *p* < 0.05 versus VDR^Δ/Δ^ baseline, °° *p* < 0.01 and ° *p* < 0.05 versus VDR^Δ/Δ^ 5RD.

**Figure 3 ijms-21-07989-f003:**
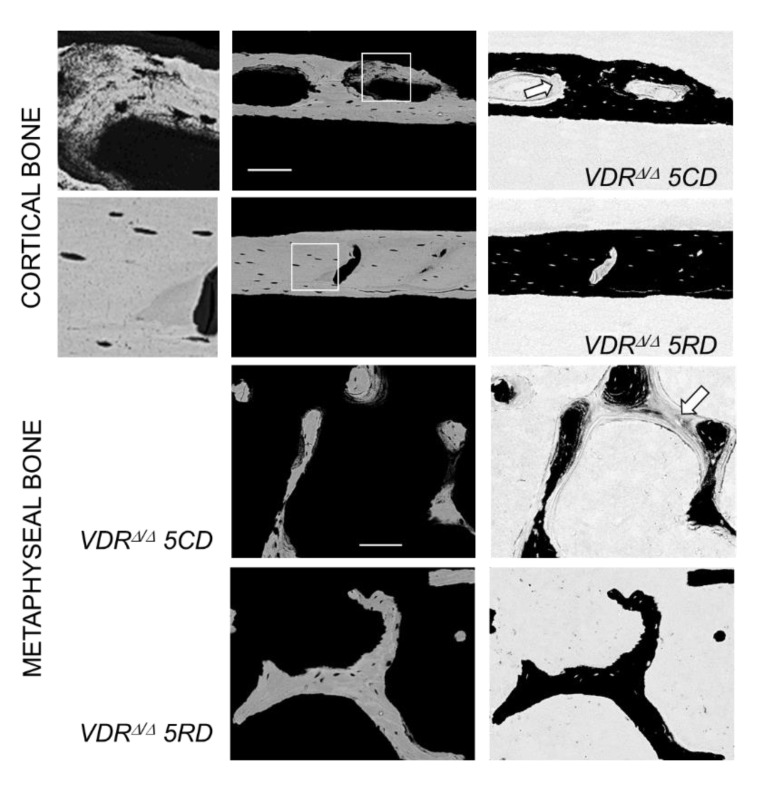
Detailed quantitative backscattered electron imaging (qBEI) images of cortical and metaphyseal spongiosa. For cortical bone, a detail of newly formed bone within a pore is shown at left; this represents a detail of the image in the middle (indicated by the white square). Noteworthy are the enlarged OLS in the newly formed bone in VRD^Δ/Δ^ 5CD in contrast to the normal OLS in the old bone of VDR^Δ/Δ^ 5RD mice. In these pictures, the brightness of the pixel is correlated with the local calcium concentration (the brighter the pixel, the higher the calcium concentration). On the right, a qBEI image with inverted contrast settings is shown. In this image, mineralized bone appears black and unmineralized osteoid appears grey. Local mineralization defects can be seen in cortical pores as well as in metaphyseal trabeculae (arrows) of the VDR^Δ/Δ^ 5CD mice, which are not present in the VDR^Δ/Δ^ 5RD mice. Scale bars indicate 100 microns.

**Figure 4 ijms-21-07989-f004:**
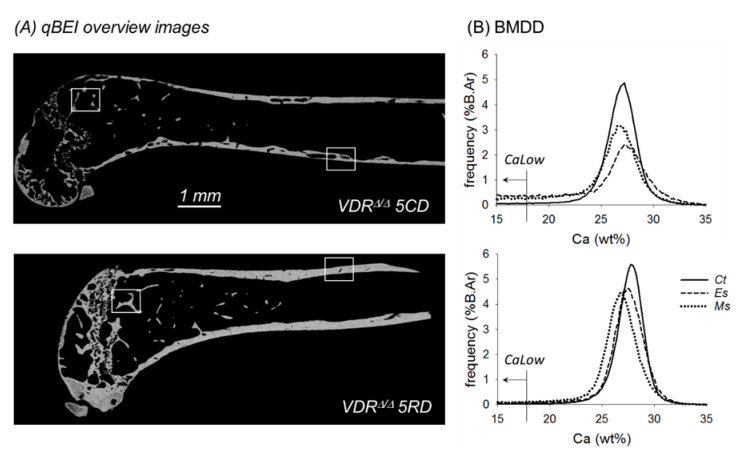
(**A**) qBEI overview images of the central longitudinal section of the femur from a VDR^Δ/Δ^ 5CD (top) and a VDR^Δ/Δ^ 5RD mouse (bottom). (**B**) Corresponding BMDD from VDR^Δ/Δ^ 5CD (top) and VDR^Δ/Δ^ 5RD (bottom). Ct., Es., and Ms.BMDD are indicated by solid, dashed, and dotted lines, respectively. Noteworthy are the differences in frequency of pixels with calcium concentrations below 17.68 wt% Ca (indicated by the arrow). This bone area is quantified by CaLow, which was highly increased in the VDR^Δ/Δ^ mice with the CD. The areas indicated by the white squares are shown in larger magnification in Figure 3.

**Table 1 ijms-21-07989-t001:** Correlation analysis of bone mineralization density distribution (BMDD) with previously obtained biochemical outcomes from all studied animals. CaMean: the average Ca concentration of the mineralized bone area; CaLow: the amount of tissue area mineralized below 17.68 wt% Ca; CaPeak: the mode/peak calcium concentration; CaWidth: the width at half-maximum of the BMDD histogram peak indicating the heterogeneity of mineralization; CaHigh: the percentage of highly mineralized bone areas (above the average 95^th^ percentile of calcium concentrations of the wild-type baseline samples); PTH: parathyroid hormone; ALP: alkaline phosphatase.

Studied Site	BMDD Parameter	PTH	ALP	Serum Ca
Cortical diaphysis	CaMean	ns	ns	ns
	CaPeak	ns	ns	ns
	CaWidth	ns	ns	ns
	CaLow	0.62 ***	0.63 ***	−0.69 ***
	CaHigh	ns	ns	ns
Epiphyseal spongiosa	CaMean	−0.72 ***	−0.73 ***	0.63 ***
	CaPeak	ns	ns	ns
	CaWidth	ns	ns	ns
	CaLow	0.86 ***	0.81 ***	−0.60 ***
	CaHigh	ns	ns	ns
Metaphyseal spongiosa	CaMean	−0.63 ***	−0.60 ***	ns
	CaPeak	ns	ns	ns
	CaWidth	ns	ns	ns
	CaLow	0.82 ***	0.77 ***	−0.66 ***
	CaHigh	ns	ns	ns

Data show Spearman rank order (SRO) correlation coefficients for significant correlations. *** *p* < 0.001 (only results with *p*-values < 0.001 were considered significant). SRO coefficients > 0 indicate that both tested variables are increasing, while SRO coefficients < 0 indicate that one variable is decreasing while the other variable is increasing. ns = not significant

**Table 2 ijms-21-07989-t002:** Mineralized bone microstructure. VDR^Δ/Δ^: mice with a non-functioning vitamin D receptor; CD: calcium challenge diet.

Studied Site	Bone Microstructure	WT BL (*n* = 5)	VDR^Δ/Δ^ BL (*n* = 4)	WT 2CD (*n* = 6)	VDR^Δ/Δ^ 2CD (*n* = 4)
Es	md.BV/TV (%)	25.5 (19.6; 34.5)	26.5 (19.2; 27.2)	32.0 (25.0;35.0)	9.7 ** (6.3; 19.0)
	md.Tb.Th (μm)	64 (61; 77)	69 (60; 78)	78 (68; 86)	50 ** (47; 62)
	md.Tb.N (/mm)	4.0 (2.9; 4.9)	3.6 (3.1; 3.9)	4.0 (3.8; 4.2)	2.0 * (1.3; 3.0)
Ms	md.BV/TV (%)	11.5 (10.6; 18.3)	9.6 (9.0; 15.9)	11.1 (9.8; 13.6)	7.1 (3.6; 12.0)
	md.Tb.Th (μm)	41 (37; 45)	41 (37; 58)	47 (41; 59)	38 (28; 54)
	md.Tb.N (/mm)	3.0 (2.8; 4.1)	2.5 (2.2; 2.8)	2.2 (1.9; 2.6)	1.7 (1.2; 2.3)

Data are median (IQR). ** *p* < 0.01, * *p* < 0.05 versus WT of same age. Es = epiphyseal spongiosa; Ms = metaphyseal spongiosa.

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
