# Peer review of "No Role of Osteocytic Osteolysis in the Development and Recovery of the Bone Phenotype Induced by Severe Secondary Hyperparathyroidism in Vitamin D Receptor Deficient Mice"

_ijms, 2020, doi:10.3390/ijms21217989_

Round 1

Reviewer 1 Report

The roles of the VDR across calcium homeostasis are important, and also nuanced. They are important in that levels of serum calcium and its impact on bone health have profound implications for the health of the population, and especially, the aging population.

The authors present a well-balanced study that has been carefully undertaken and I only have two general comments

  1. Can the authors more clearly indicate how the current study fits into the already well-established work that has been undertaken on bone physiology in the VDR knockout mice
  2. I was unclear how old the oldest mice in the study were? Can the authors clarify if their data indicate bone phenotypes in any pronounced manner in aging mice. If these data are unavailable, can they at least speculate about possible impact?

Author Response

We thank the Reviewer for the positive comments and helpful suggestions regarding our submitted work. All text changes in the manuscript are shown in blue text.

The authors present a well-balanced study that has been carefully undertaken and I only have two general comments

  1. Can the authors more clearly indicate how the current study fits into the already well-established work that has been undertaken on bone physiology in the VDR knockout mice

Response: Our BMDD findings fit perfectly to what is already known about VDR knockout mice, which is that rescue diet is able to maintain normal mineralization and that challenge diet decreases bone matrix mineralization. Additionally, the decrease in bone volume and increase in cortical pores is known from previous works. All together shows how the severe hyperparathyroidism affects bone turnover and mineralization. What was unknown so far is that the rescue diet is able to restore bone matrix mineralization also in adult VDR mice and that neither challenge nor rescue diet have a significant influence on osteocyte lacunae characteristics.

We have added this statement to the discussion line 168-174 “In general, our BMDD findings fit perfectly to what is already known about VDR knockout mice, which is that rescue diet is able to maintain normal mineralization and that challenge diet decreases average bone matrix mineralization. Additionally, the decrease in bone volume and increase in cortical pores is known from previous works. All together shows how the sHPT affects bone turnover and mineralization. What was unknown so far is that the rescue diet is able to restore bone matrix mineralization also in adult VDRD/D mice and that neither challenge nor rescue diet have a significant influence on osteocyte lacunae characteristics.”

  1. I was unclear how old the oldest mice in the study were? Can the authors clarify if their data indicate bone phenotypes in any pronounced manner in aging mice. If these data are unavailable, can they at least speculate about possible impact?

Response: The oldest mice were 9 months old. For clarification, we added this to line 306-307:  “These latter three study groups comprised the 9-month-old animals which represent the oldest mice studied in this work.”

Premature ageing based on effects in lipid and carbohydrate metabolism has been studied in VDRD/D  mice (new ref 20). These data have shown that VDRD/D mice might not be an appropriate model for studying ageing based on the aforementioned parameters as the older VDRD/D mice have a lean phenotype. However, for our knowledge, there is no evidence that the inactivating mutation of the VDR interferes with ageing of bone properties. Therefore, we have to consider the ageing effects in control mice. Concerning bone matrix mineralization, there is no evidence that BMDD changes occur in adult mice up to 16 months (new ref 21) or even up to 18 months of age (new ref 22) (i.e. up to much older age than studied in our work). Thus, we suppose that we generally can rule out any significant influence of age on BMDD in our 6 or 9 months old mice. Additionally, no age-differences were observed in osteocyte lacunae characteristics from mice between 6 and 18 months (based on another analysis-method than we used in our work) (new ref 23).

We have included following statement in the discussing (line 196-204): “Noteworthy, no information about possible premature bone ageing in the VDR mutants is available. Some evidence of premature ageing was found based on effects in lipid and carbohydrate metabolism although the VDRD/D mice might not be an appropriate model for studying such effects as the older VDR mutants have a lean phenotype [20]. Moreover, there is no evidence that BMDD changes occur by ageing in adult wild-type mice up to much older age than studied in the present work [21, 22]. This suggests that we were able to observe dietary effects only and we can generally rule out any significant influence of age on BMDD in the present work. This is likely also the case for OLS characteristics where no age-differences were observed in mice between 6 and 18 months [23].”

Reviewer 2 Report

Manuscript by Misof et al investigates the physiological role of vitamin D receptor in the regulation of normocalcemia and bone metabolism (including bone mineralization and ostecytic osteolysis) under different feeding regiments. Although mice under challenge diet showed reduced BMDD, osteomalacia and cortical pores, authors did not find the effect of VDR deficiency on the development of osteocytic osteolysis during secondary hyperparathyroidism. The results may add to the understanding of the regulation of bone metabolism by vitamin D. However, study design have considerable limitations and conclusions are not fully supported by data.
Introduction is well written. The sentence „... it appears to lack physiological function...“ sounds incorrect and should be refined (lack pathologic effect under normal calcium supply?).
Within the Results section the authors noted that they use some measurements obtained in the previous study (Weber et al, JBMR 2009) and noted that in the submitted manuscript they further analyzed the same animals (same study design). It is not completely clear did they perform all bone measurements on previously collected specimens or they included additional animals for the current study. Although it is acceptable to use some results from previous studies (in order to reduce the number of animals according to 3R), better explanation what are the novel data obtained in the submitted study and which data are already published is needed. Results are of limited novelty, since it is known that VDR mutant mice lack the bone phenotype under RD and develop bone abnormalities under CD. Although it includes some original measurements it should be better justified. Further functional experiments including TRAP stain, osteoclastogenic cultures, tissue specific (osteocyte) mutants, osteocyte cultures or similar should be added, to increase the scientific merit and to clearly add to the already known findings.
Authors concluded that VDR mutant mice on CD maintain serum Ca level by enhanced osteoclastic resorption rather than by osteocytic osteolysis, but they did not show any evidence of enhanced osteoclast activity (in vivo or in vitro). Very often they referred to the Weber et al, JBMR 2009, so it is very confusing what they actually did in the submitted study - should be better explained. The most recent publications should be added. Several limitations are explained. Limitation of including only male mice and discussion of possible sex-related differences of bone metabolism regulation by vitamin D should be added.

Author Response

We thank the Reviewer for these comments and added some further explanation to our manuscript to clarify the open issues. Text changes in the manuscript are shown in blue text.

Manuscript by Misof et al investigates the physiological role of vitamin D receptor in the regulation of normocalcemia and bone metabolism (including bone mineralization and ostecytic osteolysis) under different feeding regiments. Although mice under challenge diet showed reduced BMDD, osteomalacia and cortical pores, authors did not find the effect of VDR deficiency on the development of osteocytic osteolysis during secondary hyperparathyroidism. The results may add to the understanding of the regulation of bone metabolism by vitamin D. However, study design have considerable limitations and conclusions are not fully supported by data.

Introduction is well written. The sentence „... it appears to lack physiological function...“ sounds incorrect and should be refined (lack pathologic effect under normal calcium supply?).

Response: We added the suggested statement. Line 69-70: “Hence, although the VDR is widely expressed in bone cells, it appears to lack a physiological function under normal calcium supply [9, 15].”

Within the Results section the authors noted that they use some measurements obtained in the previous study (Weber et al, JBMR 2009) and noted that in the submitted manuscript they further analyzed the same animals (same study design). It is not completely clear did they perform all bone measurements on previously collected specimens or they included additional animals for the current study. Although it is acceptable to use some results from previous studies (in order to reduce the number of animals according to 3R), better explanation what are the novel data obtained in the submitted study and which data are already published is needed.

Response: As obviously it was not clear what already had been measured in these mice and what we have now analyzed in bone of these mice, we explain this now in more detail in the Methods section. In the former study by Weber et al. (ref 9), the focus was on the function and proliferation of parathyroid gland cells. The sHPT which occurred after switching the mice to CD was found associated with hypertrophy and hyperplasia of the parathyroid glands. Moreover, the observations in bone (bone markers, bone mineral density, and von Kossa staining of longitudinal sections) were rather a by-product in the former work in order to show the skeletal effects of the sHPT. The bone samples from the mice studied by Weber et al. (ref 9) were used now in order to study bone matrix mineralization and OLS-characteristics as described in our manuscript.

We added the following statement, line 291-296: “These femora were harvested from mice which were primarily studied for the function and proliferation of parathyroid gland cells under different diets [9]. The bone serum parameters measured in the latter previous work [9] were used for correlation analyses with bone matrix mineralization and osteocyte lacunae characteristics outcomes in the current work. The generation of the VDRD/D mice and the design of the present study using the adult onset of sHPT was described elsewhere [3, 9].”

Results are of limited novelty, since it is known that VDR mutant mice lack the bone phenotype under RD and develop bone abnormalities under CD. Although it includes some original measurements it should be better justified.

Response:  It is well known that RD is able to preserve a normal skeletal phenotype when fed after weaning. Thus, we agree, the normal bone found at 4 months old (baseline) VDRD/D mice is what we expected. However, neither the development of the bone mineralization phenotype after CD nor its rescue after RD has been described in such an adult sHPT onset model in any work before. In particular, it is very interesting that no increased osteoid volume or thickness could be observed after RD. This indicates that the collagen matrix with defective mineralization is able to mineralize to normal level during RD. This information is new as in other works where RD was fed from the beginning of life, the development of defective mineralization was prevented by RD. Thus, the situation in these growing mice on RD is very different from our mouse model.

Further functional experiments including TRAP stain, osteoclastogenic cultures, tissue specific (osteocyte) mutants, osteocyte cultures or similar should be added, to increase the scientific merit and to clearly add to the already known findings. Authors concluded that VDR mutant mice on CD maintain serum Ca level by enhanced osteoclastic resorption rather than by osteocytic osteolysis, but they did not show any evidence of enhanced osteoclast activity (in vivo or in vitro). Very often they referred to the Weber et al, JBMR 2009, so it is very confusing what they actually did in the submitted study - should be better explained. The most recent publications should be added.

Response: The increased osteoclastic activity can already be observed in the qBEI images, as in the large pores resorption lacunae are clearly visible. It should be also mentioned that murine bone does not contain Haversian channels (osteons and osteonal remodeling, respectively) under normal condition and the presence of tunneling resorption per se is therefore an indication for the increased osteoclastic resorption. Moreover, previously measured bone markers such as urinary deoxypyridinoline/creatinine excretion (ref 9) support our interpretation of the findings in the cortical qBEI images.

We added (also in Reponse to Reviewer #3) the following, line 237-245: “Although we are not able to rule out that demineralization processes (independent of the osteocytic activities) might also contribute to the observed cortical pores and defects in mineralization, there is evidence that osteoclastic resorption is responsible for the observed features. The qBEI and inverted qBEI images clearly demonstrate eroded surfaces corresponding to resorption lacunae within the pores and that the pores are only partly filled with osteoid. Both would not be seen in a general demineralization of bone, and are pointing toward osteoclastic resorption as the fundamental process leading to the large pores. Noteworthy, the increased urinary deoxypyridinoline/creatinine excretion reported in the previous work by Weber et al. [9] supports this assumption.“

It should be mentioned further that from the previous study, only bone samples have been harvested, thus neither osteoclast nor osteocyte cultures are available for these mice. We refer to the previous study for better understanding of the mineralization data of the present study. Moreover, as identical animals were analyzed in both previous and current studies, we felt that it is justified to use important information (serum parameters) from the previous work for correlation analysis with the present data. However, the use of previously obtained serum data for correlation analysis is clearly indicated in the manuscript.  No other data from the previous work were used here.

Moreover, as suggested we added further recent studies to our text, these are new ref 4 (Fleet 2017), ref 5 (Christakos et al. 2019), and ref 19 (Ryan et al. 2016).

Several limitations are explained. Limitation of including only male mice and discussion of possible sex-related differences of bone metabolism regulation by vitamin D should be added.

Response: We added the following statement about the limitation of including only male mice in our study.

Lines 273 to 277: “Moreover, we included only male mice which might have prevented us from the observation of possible sex-related differences in bone metabolism regulation by the VDR. Sex-related differences in bone microstructure and formation have been reported for VDR knockout mice fed with RD compared to control animals [19].”

Reviewer 3 Report

Misof et al. present an interesting manuscript regarding the role of osteocytic osteolysis during secondary hyperparathyroidism in Vitamin D receptor-deficient mice. The manuscript is well written and designed, the methods are appropriate, and the conclusions are supported by the data. I have a few comments which should be addressed by the authors as indicated:

- The authors are well experienced in the BMDD analysis and have done a good job in applying this technique to the different groups of this study

- It is reasonable (and previously described in younger Vdr-/- mice) that the rescue diet leads to an increase in mineralization measured via BMDD

- as one would rather expect enlarged lacunae - do the authors have an explanation for a lower OLS area in Vdr-/- mice versus WT (-12.1%, see Fig. 1 B)?

- The results are interesting based on the assumption that “calcium stress” (such as in lactation, vitamin D deficiency) has been reported to lead to osteomalacia accompanied by changes in the lacunocanalicular system (i.e., osteocytic osteolysis)

- I agree with the authors’ explanation on the differences of their findings compared to previous studies (i.e., older age and normal baseline values of Vdr-/- mice at 4 months). I could only imaging that the hypocalcemic challenge was just strong enough to lead to osteoclastic resorption but not osteocytic osteolysis (although this again does not quite fit to the strong secondary hyperparathyroidism)

- The “large cortical pores” described by the authors are most likely full of osteoid, here the weakness of the backscattered electron imaging analysis is that unmineralized bone (osteoid) cannot be mapped or quantified reliably as opposed to undecalcified histology and for example von Kossa staining. As osteoclasts most likely cannot resorb osteoid - can the authors comment on potential mechanisms of demineralization (rather than osteoclastic bone resorption)?

Author Response

We thank the Reviewer for the positive comments and helpful suggestions regarding our submitted work. All text changes in the manuscript are shown in blue text.

- It is reasonable (and previously described in younger Vdr-/- mice) that the rescue diet leads to an increase in mineralization measured via BMDD

Response: We agree that the rescue diet was able to keep BMDD at normal levels also in the younger mice as seen in the baseline levels of BMDD. This maintenance of bone mineralization at normal level during feeding of rescue diet has also been described for VDR knockout mice, see text changes line 189-191: “This finding …. and it is in line with previous observations of VDR knockout mice fed with rescue diet [14]. RD prevented the development of osteomalacia in other VDR knockout mice [19] and moreover, ….”

- as one would rather expect enlarged lacunae - do the authors have an explanation for a lower OLS area in Vdr-/- mice versus WT (-12.1%, see Fig. 1 B)?

Response: Unfortunately, we have no explanation why the VDRD/D mice have generally smaller osteocyte lacunae. Noteworthy, only the OLS-area was different, while the difference in OLS-perimeter did not achieve statistical significance.

- The results are interesting based on the assumption that “calcium stress” (such as in lactation, vitamin D deficiency) has been reported to lead to osteomalacia accompanied by changes in the lacunocanalicular system (i.e., osteocytic osteolysis)

- I agree with the authors’ explanation on the differences of their findings compared to previous studies (i.e., older age and normal baseline values of Vdr-/- mice at 4 months). I could only imaging that the hypocalcemic challenge was just strong enough to lead to osteoclastic resorption but not osteocytic osteolysis (although this again does not quite fit to the strong secondary hyperparathyroidism)

Response: From our observations we conclude that after feeding of CD in the wild-type mice, the hypocalcemic challenge was not strong enough (serum calcium levels still remain within normal range in the wild-type animals) to provoke osteocytic osteolysis and/or osteoclastic resorption. However, the lack of osteocytic osteolysis despite the severe hypocalcemia, hyperparathyroidism and osteoclastic resorption after CD in VDRD/D mice might indicate that the inactivated VDR hampers osteocytic osteolysis.

- The “large cortical pores” described by the authors are most likely full of osteoid, here the weakness of the backscattered electron imaging analysis is that unmineralized bone (osteoid) cannot be mapped or quantified reliably as opposed to undecalcified histology and for example von Kossa staining. As osteoclasts most likely cannot resorb osteoid - can the authors comment on potential mechanisms of demineralization (rather than osteoclastic bone resorption)?

Response: The inverted qBEI images (Fig. 2) as well as the von Kossa staining in the previous work (ref 9) do not give evidence for completely osteoid filled cortical pores in VDRD/D after CD. Moreover, the qBEI images show clearly the resorption lacunae within the cortical pores. Thus, for cortical bone it seems very likely that osteoclastic resorption was the origin of the large pores.

The fundamental process causing the increased osteoid volume in trabecular bone, i.e. the thick osteoid layer covering the trabeculae is less clear. We agree with the Reviewer that principally a demineralization of the bone matrix cannot be distinguished from a defective mineralization of newly formed bone in this case. However, the evidence for the latter in cortical bone together with the increased alkaline phosphatase as observed in the previous work (ref 9) point toward relatively high bone formation followed by defective mineralization.

We have added the following text to the discussion:

line 237-245: ” Although we are not able to rule out that demineralization processes (independent of the osteocytic activities) might also contribute to the observed cortical pores and defects in mineralization, there is evidence that osteoclastic resorption is responsible for the observed features. The qBEI and inverted qBEI images clearly demonstrate eroded surfaces corresponding to resorption lacunae within the pores and that the pores are only partly filled with osteoid. Both would not be seen in a general demineralization of bone, and are pointing toward osteoclastic resorption as the fundamental process leading to the large pores. Noteworthy, the increased urinary deoxypyridinoline/creatinine excretion reported in the previous work by Weber et al. [9] supports this assumption. ”

line 249-253: “We suppose also that the observed thick osteoid layer covering the mineralized trabeculae in epiphyseal and metaphyseal bone originates from defective mineralization of newly formed bone and not from demineralization. In particular, the increased alkaline phosphatase as observed in the previous work [9] points toward relatively high bone formation followed by defective mineralization.”

Round 2

Reviewer 2 Report

Authors introduced text corrections according to my comments. The manuscript text is now well corrected and appropriate. They additionally described the novelty. However, they are using bone specimens from previous study (published in 2009) and are no table to provide any additional experiments.